# Biomaterial Promotes Triboelectric Nanogenerator for Health Diagnostics and Clinical Application

**DOI:** 10.3390/nano14231885

**Published:** 2024-11-23

**Authors:** Qiliang Zhu, Enqi Sun, Yuchen Sun, Xia Cao, Ning Wang

**Affiliations:** 1Center for Green Innovation, School of Mathematics and Physics, University of Science and Technology Beijing, Beijing 100083, China; d202110424@xs.ustb.edu.cn (Q.Z.); d202310453@xs.ustb.edu.cn (E.S.); d202410475@xs.ustb.edu.cn (Y.S.); 2Beijing Institute of Nanoenergy and Nanosystems, Chinese Academy of Sciences, Beijing 100083, China; 3School of Chemistry and Biological Engineering, University of Science and Technology Beijing, Beijing 100083, China

**Keywords:** biomaterials, triboelectric nanogenerator, bionic, healthcare, self-powered devices

## Abstract

With the growing demand for personalized healthcare services, biomaterial-based triboelectric nanogenerators (BM-TENGs) have gained widespread attention due to their non-toxicity, biocompatibility, and biodegradability. This review systematically examines the working principles, material choices, biomimetic designs, and clinical application scenarios of BM-TENGs, with a focus on the use of natural biomaterials, biocomposites, hydrogels, and other materials in health diagnostics. Biomaterials show significant potential in enhancing TENG performance, improving device flexibility, and expanding application ranges, especially in early disease detection, health monitoring, and self-powered sensing devices. This paper also addresses the current challenges faced by BM-TENG technology, including performance optimization, biocompatibility, and device durability. By integrating existing research and technological advancements, this review aims to deeply analyze the development of BM-TENG technology, propose corresponding solutions, and explore its practical application prospects in the medical field.

## 1. Introduction

With the rapid development of living standards, there is an increasing demand for personalized medical services [1,2,3,4,5]. Owing to their benefits such as non-toxicity, biocompatibility, and biodegradability, biomaterial-based triboelectric nanogenerators (BM-TENGs) have emerged as a crucial technology to address this demand [6,7,8]. Compared to traditional energy-converting devices, TENG devices that are based on biological materials can not only efficiently harvest mechanical energy but also be flexibly designed to meet the needs of various medical applications [9,10,11,12]. Bioinspired design plays a crucial role in this process. By mimicking the structures and functions of biological systems in nature, bioinspired TENG devices have made significant progress in material selection, surface structure, and electrical performance optimization [13]. For example, many bioinspired materials replicate the mechanical properties of skin, muscles, or cells, giving TENGs better flexibility and durability, which demonstrates great potential, especially in personalized medical devices [14].

In the field of personalized healthcare, TENG devices provide self-powered and wearable solutions for applications like early disease detection, chronic disease management, and health monitoring [15,16,17,18,19]. For instance, wearable and implantable TENGs have garnered wide attention due to their efficiency and real-time capability in physiological signal collection [20,21,22,23,24]. As a cutting-edge energy harvesting technology, TENGs show immense potential in wearable devices. Since 2012, TENGs have attracted wide attention due to the diverse material choices and sustainability they offer [25]. Particularly driven by bioinspiration, the electrical performance of bioinspired surface structures and materials has significantly enhanced the energy conversion efficiency in bio-based applications [26]. TENGs can convert various forms of mechanical energy from daily activities (e.g., walking, breathing) into electricity [27,28,29]. Particularly in bio-based applications, TENGs have significantly improved energy conversion efficiency by optimizing the surface structure and electrical properties of biological materials [30,31].

As a core component of these devices, bio-materials not only enhance the flexibility and stretchability of TENGs but also offer excellent biocompatibility and degradability, making them suitable for long-term implantation and wearing [32,33,34]. BM-TENGs can convert mechanical energy generated by body movements and physiological activities (e.g., breathing, heartbeat, pulse) into electricity, enabling continuous, self-powered health monitoring [35,36]. Alongside natural biomaterials and biocomposites, BM-TENGs integrate a diverse range of materials such as polymers (e.g., polyurethane, PDMS), textiles (e.g., cotton, silk), gels (e.g., hydrogels, organogels), and solid–liquid composite materials (e.g., polymer composites, metal–organic frameworks). These materials are selected for their exceptional biocompatibility, mechanical properties, and sustainability, collectively enhancing the performance and expanding the application potential of BM-TENGs in personalized healthcare scenarios.

In the era of the Internet of Things (IoT), personalized BM-TENG devices can also generate real-time health data while harvesting mechanical energy from the human body and be used as self-powered sensors for diagnosis and treatment [37,38,39]. To achieve this, BM-TENG devices need to integrate sensing and therapeutic functions, creating intelligent and miniaturized health monitoring platforms for addressing a wide range of medical needs [40,41]. However, current BM-TENG devices face challenges in terms of performance, biocompatibility, durability, and sustainability [42,43]. For instance, these devices need to efficiently collect energy from the environment (such as mechanical or thermal energy) to be self-powered, but improving both the performance and biocompatibility, as well as degradation properties, remains a key issue [44,45,46].

To date, BM-TENG research has covered multiple aspects, including mechanism exploration [25], device development [47], performance optimization [48], and clinical applications [21]. Utilizing the energy and simultaneously monitoring biomechanical movements of the human body is a critical approach to achieving self-powered medical devices [49]. The output signals from TENGs can be used to monitor biochemical reactions, evolving into self-powered medical devices for disease diagnosis [50], sports analysis [51], and antibacterial/antimite applications [52].

This review comprehensively summarizes the latest advances and practical challenges of BM-TENGs and the as-based intelligent devices in personalized healthcare. It begins with the selection of biological materials, bioinspired designs, and their specific application scenarios in TENG devices, such as the regulation and monitoring of signals like respiratory rate, muscle nerve stimulation, bacterial detection, and sterilization and mite removal functions. Key characteristics of BM-TENGs, including stretchability, shape adaptability, breathability, and self-healing capability, play a pivotal role in enhancing their functional performance in these applications. We then provide an in-depth analysis of the potential of BM-TENGs as self-powered energy sources, particularly their prospects in wearable and implantable devices, and discuss design strategies and functional integration across various application scenarios, including respiratory and cardiovascular monitoring, neuromuscular diseases diagnosis, and bacterial diagnostics. Additionally, this paper discusses various biomaterials, including natural biomaterials, biocomposites, and hydrogels, from the perspectives of biocompatibility and sustainability, to enhance TENG output performance and expand application domains. Moreover, we address the challenges and future directions of BM-TENG development, emphasizing the critical role of interdisciplinary collaboration in advancing BM-TENG technology towards more efficient energy utilization and broader clinical applications (Figure 1).

## 2. Energy Collection

TENG has emerged during a period marked by increasing demand for clinical applications, proving itself as a dependable and sustainable energy harvester. It offers advantages such as compact size [41], light weight [66], high efficiency [67], low cost [68], and portability [69]. TENG can harness various forms of mechanical energy from the environment, primarily through the synergistic effects of the triboelectric effect and electrostatic induction, converting it into electrical energy to power medical diagnostic systems [70,71]. However, achieving practical applications requires TENG designs that strike a balance between comfort, durability, and, notably, high-output performance. This aspect is crucial for realizing the full potential of self-powered medical diagnostic systems [72,73].

### 2.1. Working Principle

Triboelectric charging is a common phenomenon encountered in daily life [74,75]. It occurs when the surfaces of two electrodes come into contact, causing electron clouds within the materials to overlap and generate triboelectricity [76]. The origin of the triboelectric effect lies in the differences in electron affinity between materials, which influences their ability to gain or lose electrons during contact. This phenomenon can be applied in the medical field, particularly in self-powered biosensors and implantable devices. These devices can harvest biomechanical energy from body movements or mechanical stimuli and convert it into electrical energy, supporting the operation and data transmission of various biomedical sensors. Upon separation of the triboelectric surfaces without external force, an internal potential difference is established within the device [77]. This prompts carriers to flow from one electrode to the other, counteracting the potential change induced by electrostatic induction and resulting in a positive current [78]. When an external force is applied and the electrodes of the device move relative to each other, carriers within the material flow in the opposite direction, thereby altering the polarity of the potential [79]. The efficient conversion of mechanical energy to electrical energy in TENGs is closely linked to the effective contact between the materials, which can be enhanced by surface roughness. Moreover, when these two electrodes repeatedly come into contact, this process cycles, resulting in periodic energy output [80].

TENGs typically operate in four primary modes: vertical contact–separation, sliding, single-electrode, and freestanding triboelectric-layer modes [76,81,82].

The most commonly utilized mode in TENGs is the vertical contact–separation mode (Figure 2a), where one electrode is attached to the top and the other to the bottom of a triboelectric layer. As these electrodes move vertically relative to each other under an external force, charges are generated, leading to a fluctuating potential between them during contact or separation, thereby generating current in the external circuit [27,83]. This characteristic can be applied to medical diagnostic devices to monitor body signals, such as pulse and muscle activity.

Following this is the sliding mode (Figure 2b), which resembles the vertical contact–separation mode. Here, a force parallel to the interface induces relative displacement between the device’s electrodes, generating a dense accumulation of triboelectric charges. The quantity of these charges varies periodically with changes in the contact area [39,84], making it suitable for dynamic medical applications, such as continuous patient condition monitoring.

The single-electrode mode (Figure 2c) offers an advantage due to its compatibility with irregular movements. In this mode, only one electrode needs grounding as a reference, facilitating charge generation upon contact with a freely moving object [85,86], making it highly suitable for portable medical devices by offering versatility and portability.

In the freestanding triboelectric-layer mode (Figure 2d), a freely moving component is integrated, altering the potential distribution asymmetrically as it shifts position. This movement prompts a continuous flow of electrons between the electrodes, resulting in periodic energy output, making it ideal for wearable or implantable medical devices [87,88].

These various TENG working modes can provide extensive support for future medical technologies, from real-time monitoring to self-powered systems in long-term implants, enhancing the intelligence and self-sufficiency of medical devices.

### 2.2. Wearable BM-TENGs

To meet the practical operational needs of self-powered portable electronic devices, various wearable TENGs have been designed to collect biomechanical energy for external use. The electricity generated by these TENGs can power small electronic devices such as LED lights, watches, and capacitors. Furthermore, wearable TENGs can also capture physiological data such as pulse, heart rate, and respiratory rate through the electrical signals they generate [89]. These devices primarily function outside the body, providing a practical solution for powering external devices while also monitoring vital signs.

Hao et al. developed a high-output multilayer-structured triboelectric nanogenerator (HM-TENG) that efficiently harvested human mechanical energy by using a nylon/L-cystine composite nanofiber film (NCNF) (Figure 3a) [90]. They found that adding 8% L-cystine to an NCNF significantly improved the output performance of the TENG, with the open-circuit voltage, short-circuit current, and short-circuit transfer charge being 2.98, 5.33, and 3.41 times higher than those of TENGs made from pure nylon nanofiber film. The team also designed a power management circuit and successfully installed the HM-TENG on the foot for energy harvesting, powering various self-powered wearable devices such as hygrometers, pedometers, and sports watches. This research not only provides a novel method for developing high-output TENGs but also offers an important solution for designing self-powered electronic devices, demonstrating the broad application prospects for using human mechanical energy to drive portable and wearable devices.

In separate research, Wu et al. developed a nanogap TENG based on a biodegradable polybutylene succinate (PBS) film (Figure 3b) [91]. Using biocompatible silicone rubber (SR) as the negative counterpart and comparing it with other biodegradable plastics like poly (lactic acid) (PLA) and polycaprolactone (PCL), they found that the PBS-based TENG had the highest output performance, up to 3.5 times higher than that of other TENGs made from biodegradable materials. This performance enhancement was attributed to the rough surface of PBS and its high ester group content. The TENG showed potential as a pressure and angle sensor for health monitoring, and its environmental stability in air, natural water, and phosphate-buffered saline was verified. Moreover, the antibacterial properties of the PBS film indicated promising applications in wearable and implantable electronic devices. The study underscored the potential of biocompatible and environmentally stable TENGs for integration into wearable electronics and biomedical systems.

### 2.3. Implantable BM-TENGs

Implantable electronic devices are crucial medical technologies for monitoring physiological responses within the body. TENGs, as power sources for implantable medical devices, have greatly facilitated the application of physiological signal sensors, smart stomachs, pacemakers, cochlear implants, and deep brain stimulators. These implantable TENGs provide a reliable and biocompatible energy source for devices that operate inside the body, facilitating continuous health monitoring and therapeutic interventions.

In one study, Niu et al. emphasized the development of a silk nanoribbon (SNR)-based bio-triboelectric nanogenerator (bio-TENG) tailored for implantable self-powered electronic devices (Figure 3c) [64]. The study achieved high biocompatibility, biodegradability, and a controllable degradation rate using a bio-TENG made entirely of silk and magnesium (Mg), thus avoiding the need for secondary surgery and minimizing the risk of inflammation. By directly exfoliating SNRs from natural silk at a thickness of 0.38 nm and maintaining its original meso/nanoscale structure, the team combined it with regenerated silk fibroin film (RSFF) as a friction layer. By controlling surface roughness and structural differences, the device achieved a maximum output voltage of 41.64 V, current of 0.5 μA, and power density of 86.7 mW/m^2^. The TENG also demonstrated high sensitivity, capable of detecting light forces such as finger tapping, footsteps, elbow movement, and even human pulse, indicating its broad application prospects in implantable bioelectronic devices, especially pacemakers and implantable sensors.

In another study, Kim and colleagues developed a TENG utilizing biocompatible and biodegradable hydrogel membranes derived from hyaluronic acid (HA) [92]. This innovative HA-TENG efficiently converted mechanical energy into electrical energy, serving as a sustainable power source for bio-implantable devices (Figure 3d). The device demonstrated impressive power output, long-term stability, and low cytotoxicity, making it suitable for applications in tissue engineering and various biomedical fields. Fabricated with pure HA and crosslinked HA hydrogel membranes, the HA-TENG consistently generated energy over extended testing periods. Its successful implementation represents a compelling proof-of-concept for biodegradable, implantable devices, paving the way for transient self-powered systems across diverse biomedical applications.

Furthermore, to minimize the invasiveness of device implantation surgeries, Zheng et al. developed a biodegradable TENG designed to harness energy from biomechanical motions such as heartbeats [46]. The device utilized biodegradable materials, comprising two chosen BDP layers (PLGA, PVA, PCL, and PHB/V), with a 200 μm BDP spacer between the friction layers. A 50 nm magnesium (Mg) film served as the electrode layer, positioned on one side of each friction layer. The entire structure was encapsulated with a 100 μm BDP layer to protect it from the surrounding physiological environment. Over time, the biodegradable TENG gradually degraded, reaching nearly complete mass loss after 90 days. This biodegradable feature eliminates the need for the surgical removal of cardiovascular therapeutic devices, reducing risks and improving patient quality of life.

In summary, wearable BM-TENGs focus on portability and everyday practicality, emphasizing the collection of biomechanical energy for external use. They generate electrical signals capable of capturing physiological data such as pulse, heart rate, and respiratory rate. In contrast, implantable BM-TENGs serve as energy sources for implantable medical devices, widely used for monitoring internal physiological responses. They offer stability and reliability for long-term monitoring and therapeutic applications, collectively advancing the application and development of self-powered technologies in the medical electronics field.

## 3. Recent Progress in BM-TENGs

### 3.1. Materials and Structure Used in Contact Surfaces

The performance of TENG is closely related to the triboelectric charge density of the selected materials [93,94,95]. In the field of biomaterials, natural polymers, biocomposites, hydrogels, ceramic materials, and nanomaterials have been widely applied in the design of TENG interfaces. These materials not only exhibit excellent electrical properties but also offer superior biocompatibility and environmental friendliness, making them highly suitable for biomedical applications.

Biomaterials applied in TENGs for medical purposes include a wide range of substances that ensure safe interaction with biological systems. These materials, with their combination of electrical performance and biological compatibility, enable the development of self-powered devices for health monitoring, energy harvesting, and medical implants, offering potential for both wearable and implantable applications.

#### 3.1.1. Natural BM

Most TENGs are traditionally built using inorganic materials that lack biocompatibility, limiting their utility in wearable and implantable medical devices. Lignin, the most abundant aromatic biomaterial in nature, provides structural support and enhances the biomechanical strength of plants, suggesting significant potential for its application in biomedical devices.

For example, Saqib et al. investigated the feasibility of using lignin-based waste shell fragments (WFSs) derived from almonds (As), walnuts (Ws), and pistachios (Pis) as positive triboelectric materials in TENGs (Figure 4a) [55]. Their study focused particularly on pistachio WFSs, integrating them with materials such as polytetrafluoroethylene (PTFE) and polyethylene terephthalate (PET). They found that Pi-WFSs demonstrated superior performance, exhibiting the highest open-circuit voltage, short-circuit current, and peak power density. These WFS-based TENGs were successfully applied to power medical devices, drive biosensors, and support self-powered wearable medical equipment. The research not only offers a new approach to resource recovery but also highlights the potential of natural materials in the biomedical field, especially in the development of eco-friendly, cost-effective, and efficient medical devices.

Plant proteins such as rice protein (RP), typically considered byproducts of the starch industry and often used as boiler fuel or animal feed, contribute to resource wastage. Jiang et al. explored the utilization of rice husk powder (RG), derived from these byproducts, to investigate the triboelectric behavior mechanism related to protein structure (Figure 4b) [54]. Employing pH cycle interface engineering, they demonstrated that the secondary structure of RG significantly influences its triboelectric performance. They successfully enhanced the output power density of the BM-TENG by approximately 16 times. This study not only promotes the efficient use of resources but also demonstrates the tunability of plant proteins as natural materials, particularly in the medical field for applications such as smart implants and wearable devices, showcasing the potential of sustainable biomaterials.

Sun et al. proposed a method to create a flexible, transparent, and fully sustainable fish gelatin-based TENG, using fish scales from kitchen waste to produce gelatin films as triboelectric layers [98]. These films degrade rapidly in the natural environment, providing eco-friendly properties. Introducing dopamine and fluorosilane modifications to the triboelectric layers resulted in the formation of a highly efficient triboelectric pair. The TENG device exhibited excellent output performance, which could be further enhanced through surface structuring and size optimization. This device significantly outperformed previously reported sustainable TENGs and successfully powered various electronic components. Moreover, the fish gelatin-based TENG could be applied to self-powered wearable sensors for human motion and human–machine interaction, demonstrating great potential in flexible, green medical devices, particularly for health monitoring and medical assistive equipment.

The research by Saqib, Jiang, and Sun collectively highlights the immense application potential of natural biomaterials for TENG devices, especially in the medical field. These innovative materials are expected to provide safer and more effective solutions in health monitoring, disease management, and human–machine interaction, significantly improving the sustainability of medical devices and enhancing patient experiences.

#### 3.1.2. Biocomposite BM

The application of biocomposites in TENGs demonstrates significant advantages and a broad range of potential applications. By optimizing the structure and composition of these materials, it is possible to significantly enhance the energy conversion efficiency and stability of TENGs. Additionally, the degradability and environmental friendliness of these materials highlight their enormous potential in medical and wearable devices.

For instance, Gao et al. successfully developed a high-performance and fully degradable chitosan (CS)-based TENG sensor by combining a plasticizer (glycerol) and a templating method (Figure 4c) [62]. Their research showed that the CS–glycerol friction layer significantly improved electrical generation performance, with an open-circuit voltage and power density greatly exceeding those of traditional CS-based TENGs without the template method. Furthermore, this sensor could fully degrade in soil after completing its operational cycle, providing an eco-friendly and degradable wearable sensor solution, particularly suitable for disposable medical products. This innovation showcases the immense potential of using natural materials for efficient energy conversion and information transmission, especially in medical and environmental monitoring applications.

Candido et al. developed a PVA/SF-based TENG (Figure 4d) by incorporating silk fibroin (SF) into polyvinyl alcohol using a straightforward method [53]. This impregnation process significantly impacted all polarization processes, directly influencing the device’s dielectric properties. However, pure SF does not meet commercial standards due to its inferior mechanical performance and susceptibility to degradation under conditions such as high temperatures, humidity, and extreme pH levels. In contrast, Xu and colleagues pioneered a flexible, stretchable, and fully biodegradable TENG capable of harvesting biomechanical energy both in vivo and in vitro [99]. They introduced mesoscopic doping to enhance the structural stability of regenerated SF, resulting in silk films that exhibited excellent chemical stability (withstanding temperatures of up to 100 °C and pH levels ranging from 3 to 11) and outstanding mechanical properties (approximately 250% stretchability and enduring 1000 bending cycles at a 2 cm radius). These advancements position doped silk materials as promising candidates for applications in human health.

#### 3.1.3. Hydrogel BM

Hydrogel materials, known for their high water content and excellent flexibility, have become a popular choice for designing contact surfaces in TENGs. Hydrogels based on gelatin and alginate, with their good biocompatibility and ability to adsorb bodily fluids, are particularly suitable for energy harvesting on the skin’s surface.

For instance, Yu et al. conducted an innovative study to develop a single-electrode TENG skin patch using MoS_2_ and a gelatin–methacryloyl (GelMA) hydrogel (Figure 4e) [96]. MoS_2_ exhibited excellent electrical conductivity and remarkable photothermal conversion properties, complemented by GelMA’s biocompatibility. This TENG device effectively harvested biophysical energy, generated an electric field around damaged tissues, and utilized near-infrared photothermal effects to expedite wound healing. Moreover, the TENG functioned as a real-time sensor for monitoring physiological signals. The prototype demonstrated significant open-circuit voltage and short-circuit current outputs, with in vitro experiments confirming its ability to enhance cell migration through photothermal heating and real-time electrical stimulation. Animal studies further validated its efficacy in promoting collagen deposition and angiogenesis, thereby significantly accelerating tissue regeneration and wound healing. These findings highlight its potential for self-powered wearable electronics, particularly in medical and environmental monitoring applications.

Kanokpaka et al. introduced a novel, self-healing, glucose-adaptive, hydrogel-based triboelectric biosensor (GAH-TES) for real-time glucose monitoring in human sweat (Figure 4f) [97]. The research utilized β-cyclodextrin-encapsulated glucose oxidase (GOx) to construct a glucose-adaptive PVA hydrogel, which exhibited excellent electrical conductivity in various glucose environments. Integrating this adaptive hydrogel into a TENG enabled efficient energy conversion from glucose stimuli in human sweat to electrical output. The study demonstrated that higher glucose concentrations enhanced the sensor’s electrical performance due to increased ionic strength from enzymatic activity. The GAH-TES not only possessed excellent selectivity, stability, and reproducibility but also featured LED visualization for glucose levels exceeding normal health ranges. The research highlights GAH-TES’s substantial potential in diabetes management by providing a promising, non-invasive, continuous glucose monitoring platform, significantly advancing self-powered and real-time detection technologies in medical diagnostics.

Furthermore, combining hydrogels with nanomaterials such as cellulose nanofibers or graphene further enhances TENGs’ mechanical performance and charge output, broadening their application in wearable devices and implantable biomedical sensors.

#### 3.1.4. Other BM

Ceramic materials, particularly lead zirconate titanate (PZT) piezoelectric ceramics, have garnered significant attention in biomedical devices and TENGs due to their exceptional piezoelectric properties. PZT not only exhibits a high piezoelectric constant but also maintains stable performance in complex mechanical environments, making it an ideal material for self-powered systems and multifunctional biosensors. As a result, integrating ceramic materials into self-powered systems holds tremendous potential for continuous physiological signal monitoring and advancing healthcare applications.

Zhu’s team advanced health applications by developing a self-powered intelligent sock (S2-sock) that integrates energy harvesting and physiological signal sensing [100]. Combining a PEDOT-coated TENG with a PZT piezoelectric chip, the sock collects energy from motion and detects signals like gait, contact force, and sweat levels. It generates up to 1.71 mW under light jumping, paving the way for self-sustained smart textiles. Using sensor fusion, it precisely captures physiological data, with TENG and PZT working together for rapid sweat detection. The S2-sock holds promise for smart homes, health monitoring, and personalized medicine, enhancing life quality and health management.

In addition, nanomaterials have shown great potential as bio-based materials for TENG and healthcare applications. For example, Chen et al. significantly improved the output power density of TENG by incorporating Ag@SiO_2_ core–shell nanoparticles into tribo-materials [101] utilizing the surface plasmon effect, achieving a maximum output power density of 4.375 mW/cm^2^. The design of the shell not only alleviated the leakage effect of conventional nanoparticles but also further enhanced the TENG’s output performance through the surface plasmon effect. Moreover, this study innovatively combined high-performance TENG with traditional Chinese medicine acupuncture, developing a self-powered electroacupuncture system that can rapidly regulate cardiovascular function, offering broad potential for healthcare applications.

#### 3.1.5. Unique Properties

The critical required characteristics of biomaterial-based TENGs include stretchability, shape adaptability, breathability, and self-healing capability. Compared to traditional synthetic materials, biomaterials exhibit greater flexibility, lower rigidity, and higher compatibility with human tissues, making them more suitable for wearable or implantable TENGs. Additionally, the self-healing properties of biomaterials ensure long-term reliability, while their breathability enhances user comfort, further underscoring their competitive edge in biomedical applications. These attributes not only influence their performance in medical electronic devices but also determine their wide applicability and practical feasibility in medical applications. With advancements in medical technology and increasing demand for personalized healthcare, developing biomaterial-based TENGs with these features has become a significant focus of current research.

Excellent stretchability is also a fundamental characteristic ensuring the conformity of wearable sensors with the deformation of the human body during motion. Fu et al. developed a fibrous stretchable TENG-based sensor with a core–sheath structure for physiological monitoring [58]. The sensor showcased resilience, with stable performance under significant deformations. It detected ultra-low pressures and demonstrated high sensitivity to external forces. The FS-TENG sensor was applied for monitoring human motions and physiological indicators, and a tactile sensor array was developed for real-time pressure distribution recognition, highlighting its potential in healthcare monitoring and motion sensing.

With the background of breathability, Tan et al. successfully developed an open-porous PDMS-coated fabric-based TENG (oPF-TENG) [57]. They achieved good breathability of the fabric by utilizing the synergistic effect of insoluble NaCl, DBP, and soluble silicone oil. The open-porous structure formed airflow channels with air permeability of approximately 73.3 ± 4.2 mm/s. Additionally, the porous structure and PVDF fillers enhanced the triboelectric performance of the oPF-TENG. The device exhibited stable electrical performance in energy harvesting and self-powered sensing, even under cyclic washing and repeated testing conditions. oPF-TENGs, with their stable performance even after repeated washing and testing, demonstrate clear applicability as energy-harvesting insoles and wearable sensors for monitoring body movements, supporting their potential in biomechanical energy and self-powered sensing.

The application of self-healing properties and biomaterials in flexible electronic devices is an important research direction. Jiang et al. successfully fabricated an ultra-stretchable TENG by incorporating hydrogen bonds and dynamic metal–ligand coordination into polydimethylsiloxane chains [59]. This material exhibits 100% self-healing efficiency at room temperature, capable of autonomously repairing fractures and abrasions, highlighting its potential applications as a flexible power source and self-powered pressure sensor in flexible electronic devices.

These attributes not only enhance performance but also broaden applicability and practical feasibility in medical applications. With advancements in medical technology and the rising demand for personalized healthcare, research on developing BM-TENGs endowed with these features has intensified.

#### 3.1.6. Bionic Design

Driven by advances in biomimetic technology, researchers have developed high-performance, self-powered sensor systems by drawing inspiration from the complex mechanisms of natural biological structures and functions. These systems not only mimic the efficient energy conversion and sensing processes found in nature but have also achieved significant breakthroughs in sensitivity, stability, and durability. By incorporating biomimetic structures, these materials can optimize energy harvesting and improve performance significantly. For example, imitating the microscopic structures of natural organisms such as octopus tentacles, lotus leaves, butterfly wings, and fish scales, biomimetic sensors exhibit great potential in optimizing energy harvesting and signal transmission. This biomimetic design concept effectively converts mechanical, thermal, and other biomechanical energy into electrical energy while also advancing the broad application of self-powered sensor systems in precision medicine, wearable devices, and the Internet of Things.

A notable achievement is the high-sensitivity flexible sensor based on carbonized ZIF-8@loofah (CZL) developed by Yue et al. (Figure 4g) [56]. The carbonized ZIF-8 nanoparticles simulated octopus tentacles, forming high-density contact–separation points, significantly improving sensor sensitivity. The CZL-based sensor demonstrated ultra-high sensitivity, fast response times (22 ms), and excellent durability (over 10,000 cycles), effectively monitoring complex human activities such as pulse and vocalization. Additionally, CZL-based TENGs converted irregular biomechanical energy into electrical energy, providing a sustainable power source for the continuous operation of micro-sensing systems.

In a similar line of research, Li et al. proposed a bioinspired, sweat-resistant, wearable triboelectric nanogenerator (BSRW-TENG) for movement monitoring during exercise [102]. The BSRW-TENG featured two bioinspired superhydrophobic and self-cleaning triboelectric layers (elastic resin and polydimethylsiloxane (PDMS)), incorporating hierarchical micro/nanostructures that replicated the lotus leaf. These biomimetic structures not only enhanced the functional properties of the materials but also ensured consistent performance under varying environmental conditions. This design approach doubled the output of the BSRW-TENG and provided excellent contamination and humidity resistance. After saline (0.9%) was dripped and evaporated, the BSRW-TENG’s output remained unchanged, while that of the flat-TENG decreased by 41%. Additionally, the BSRW-TENG showed only an 11% reduction in output as the relative humidity increased from 10% to 80%. The device effectively monitored exercises like dumbbell curls and running, with stable performance before and after sweating. In summary, the biocompatibility of materials and processes is a primary prerequisite for BM-TENGs in the healthcare field. Based on bioinspired structural designs, creating materials with suitable properties and developing related processes according to specific needs may garner more public recognition than merely pursuing high TENG output performance (Table 1). The integration of bionic design not only enhances material properties but also broadens the scope of potential applications in diverse fields. Only when the basic design of TENGs meets these standards can we truly explore more application scenarios and the practical value.

### 3.2. Self-Powered Systems for Respiratory and Cardiovascular Monitoring

#### 3.2.1. Respiratory Real-Time Diagnosis

Human respiration is considered a key indicator for real-time health monitoring and the early detection of heart failure and sleep apnea. However, most respiratory sensors are expensive, immobile, or require rechargeable batteries, limiting their mobility and daily use. Consequently, bio-friendly TENGs have been widely used for real-time respiratory monitoring.

For instance, Rajabi et al. enhanced the performance of cellulose nanofiber (CNF)-based TENGs by incorporating diatomaceous silica (DFs) (Figure 5a) [65]. The hierarchical porous 3D structure and large surface area of DF, combined with CNF materials and structures, enhanced electron transfer and increased the surface roughness of the DF-CNF composite film. This composite film exhibited high mechanical strength and electron-rich surfaces and was low-cost, making it ideal for TENGs. The biocompatibility of this TENG was validated through cytotoxicity and animal tests. It was used in smart masks that successfully monitored various breathing patterns, providing new possibilities for future TENG applications in health monitoring.

Similarly, during the COVID-19 pandemic, a smart mask based on an all-fabric TENG (AF-TENG) was designed to monitor breathing and improve medical applications (Figure 5b) [103]. This mask used ultra-high-molecular-weight polyethylene (UHMWPE) and cotton fabrics as negative and positive triboelectric layers, offering eco-friendly breathing monitoring and early warning features. The AF-TENG mask monitored respiration in real time using an Arduino microcontroller system, providing health monitoring for patients and the elderly, especially during pandemics.

Tan et al. developed a high-performance, biocompatible porous TENG using a composite of SF and MXene aerogel (SF@MXene-A) (Figure 5c) [104]. The positive triboelectric layer was made of SF@MXene-A, while the negative layer used a PDMS sponge to increase surface area and charge density. The breathable TENG mask generated 1.96 V from breathing, showing potential for monitoring respiratory diseases like asthma.

Overall, BM-TENGs show great potential in respiratory disease monitoring. However, further studies are required to validate the accuracy and sensitivity of TENG monitoring devices in clinical applications, particularly under varying breathing patterns and complex environments.

#### 3.2.2. Cardiac Real-Time Diagnosis

The continuous operation of implantable bioelectronic devices can reduce the economic burden and health risks associated with surgeries for device removal or replacement. TENGs are a promising option for low-power implantable electronics.

An outstanding example of a modular BM-TENG for real-time cardiac diagnosis is the multifunctional SF-based porous scaffold designed by Tufan et al. (Figure 5d) [61]. This scaffold combined the capabilities of cardiomyocyte differentiation and energy harvesting. By mimicking cardiac motion, it used induced pluripotent stem cells (iPSCs) to generate cardiomyocytes while converting mechanical energy into electrical energy. The SF scaffold featured optimized pore size (379 ± 34 μm), porosity (79 ± 1%), and pore interconnectivity (67 ± 1%), crucial for cardiomyocyte differentiation. Adding carbon nanofibers (CNF) enhanced the scaffold’s elasticity and conductivity, making it suitable for both cardiac tissue regeneration and as an electrode for TENGs to harvest energy from heartbeats. The SF/CNF scaffold also had excellent biocompatibility and mechanical strength and appropriate degradation properties, ensuring long-term stability in vivo.

Jiang et al. developed a TENG device utilizing natural bioabsorbable polymers (NBPs) including cellulose, chitin, rice protein (RP), SF, and egg white (EW). SF was chosen as the encapsulation layer for its triboelectric properties and resistance to body fluids following methanol treatment [20]. Two NBPs were used as triboelectric layers, with an ultra-thin magnesium film as the back electrode. ICP etching created nanostructures on the surface, enhancing contact area and charge generation. The device materials were biodegradable, with tunable degradation times. In rat experiments, methanol-treated SF showed slower degradation, lasting 42 days, while untreated SF dissolved in 21 days. The device demonstrated good biocompatibility, with no inflammation observed after degradation. By employing TENG as a power source, the contraction rates of impaired cardiomyocyte clusters were enhanced, and the uniformity of their contractions was improved. This approach offers a novel and effective solution for treating heart conditions like bradycardia and arrhythmia and holds promise for both diagnostic and therapeutic applications in cardiology.

In conclusion, extensive research supports that BM-TENGs can harness the energy from heartbeats to power pacemakers. In vitro and in vivo experiments have consistently shown substantial voltage outputs, sufficient for continuous energy harvesting from natural heart motion via BM-TENGs. Device optimization has endowed BM-TENGs with promising features, including high energy output and biodegradability, enabling a symbiotic energy cycle within the body. This underscores the great potential of energy-harvesting devices in the development of personalized healthcare.

#### 3.2.3. Pulse Real-Time Diagnosis

Pulse waveform analysis offers valuable clinical insights for diagnosing cardiovascular diseases, encompassing parameters such as heart rate (HR), pulse wave velocity (PWV), and blood pressure (BP). Critical conditions like arrhythmia, coronary artery disease, and hypertension play pivotal roles in cardiovascular diagnosis. By analyzing pulse waves, relevant information can be obtained, simplifying the process of cardiovascular disease monitoring. BM-TENG-based pulse wave monitoring sensors address the limitations of traditional electronic devices. Scientists are currently developing wearable TENGs for pulse wave monitoring, offering accurate, non-invasive, and long-term pulse measurements.

A study by Lou et al. involved a full-fiber structure TENG pressure sensor made using electrospinning technology (Figure 5e) [105]. This sensor was composed of a polyvinylidene fluoride/Ag nanowire membrane, ethyl cellulose membrane, and two layers of conductive fabric, operated in vertical contact–separation mode. By introducing a layered rough structure on the nanofibers, the sensor’s adaptability was enhanced. The experimental results showed sensitivities of 1.67 and 0.2 mV Pa^−1^ in the 0–3 and 3–32 kPa pressure ranges, respectively, and the sensor maintained good mechanical stability after 7200 cycles. Researchers placed the sensor on a healthy woman’s neck, successfully capturing clear pulse waveforms. This textile-based pressure sensor offers an effective tool for long-term cardiovascular disease monitoring.

### 3.3. Neuromuscular Self-Powered System Design

#### 3.3.1. Exercise Monitoring and Analysis

Soft tissues in the constantly moving human body, like muscles, tendons, and ligaments, are prone to injury during intense outdoor activities [106,107]. Without proper injury monitoring, the healing process may be hindered, potentially leading to impaired bodily functions in severe cases. As a result, it is crucial to have sensors that can quickly, affordably, and easily monitor physical parameters such as pressure and strain on muscles and ligaments. This allows for the development of personalized treatments tailored to specific injuries, while the ongoing observation of tissue repair is vital for effective soft tissue rehabilitation.

Bincy et al. used arrowroot, corn, potato, and cassava starch as active layers to fabricate a TENG (Figure 6a) [108]. By using density functional theory (DFT) to analyze triboelectric active sites, it was found that the corn starch-based TENG (CS-TENG) performed best. The CS-TENG was used to construct a green, wearable IoT system, combining mechanical–electrical sensors and a wireless personal health monitoring system (WPHM). It can track physical activities and calculate calorie consumption. Supported by an Android app, the system allows real-time monitoring of various exercises, such as grip strength and running.

Sheng et al. developed an organic gel/silicone fiber helical sensor-based TENG (OFS-TENG) for implantable ligament strain monitoring without an external power source, mainly targeting muscle and ligament injuries caused by sports or disease (Figure 6b) [63]. In contrast to existing implantable sensors that require external power or lack flexibility and stability, OFS-TENG exhibited high stability and extreme stretchability (up to 600%). The sensor, comprising organic gel fibers and silicone fibers in a double-helix structure, offers rapid preparation (15 s), high transparency (>95%), and exceptional stability (over six months). The OFS-TENG was successfully implanted into a rabbit’s patellar ligament, allowing for the real-time monitoring of ligament stretching and muscle strain. This provides an effective solution for the real-time diagnosis of muscle and ligament injuries. This self-powered OFS-TENG sensor can collect real-time data related to human muscles and ligaments, making it suitable for implantable medical devices.

Overall, the studies by Bincy et al. and Sheng et al. provide innovative technological solutions for monitoring human soft tissues. These works have laid a solid foundation for the development of flexible, wearable devices and implantable medical sensors, with broad application prospects. In particular, they demonstrate the ability to track movement and energy consumption in real time, making them especially promising in the fields of fitness and health monitoring.

#### 3.3.2. Parkinson’s Disease Diagnosis

A typical example of a self-powered system was designed by Kim et al. (Figure 6c) [60]. This system integrates a highly stretchable and self-healing TENG with two modules: a catechol–chitosan–diatom hydrogel (CCDHG) electrode module and an M-shaped Kapton film module. The CCDHG electrode module provides excellent biocompatibility, stretchability, self-healing ability, and conductivity. In contrast, the M-shaped Kapton film module enhances the contact area and sensitivity of the TENG, serving as a tremor sensor to detect low-frequency vibrations, which is crucial for Parkinson’s disease diagnosis and biomedical health monitoring. Moreover, this self-powered tremor sensor shows potential applications in smart e-skin, soft robotics, and wearable bioelectronic devices.

Another example comes from Wang et al. (Figure 6d) [109], who developed a device composed of two main modules: a flexible strain sensor and a data processing module. The flexible strain sensor was crafted from a graphene oxide–polyacrylamide (GO-PAM) hydrogel, tailored for clinical applications with a notable focus on detecting Parkinson’s disease and hemiplegia. This self-powered strain sensor, leveraging GO-PAM hydrogel, demonstrates exceptional sensitivity in detecting subtle body movements, including gait. Integrated into an in-shoe monitoring system with an artificial neural network algorithm, it accurately identifies speed variations and pathological changes in daily activities, providing a convenient and efficient solution for early diagnosis, rehabilitation assessment, and patient treatment.

### 3.4. BM-TENG Systems for Bacterial Diagnostics and Sterilization

#### 3.4.1. Bacterial Diagnostics

TENGs based on biomaterials can convert mechanical energy into electrical energy, offering a stable power source for bacterial diagnosis. Their sensitive electrical signal response characteristics allow for the real-time detection of bacterial attachment and growth, as well as the analysis of bacterial colonies and biofilm formation. Coupled with sensor technology, TENGs can achieve rapid and convenient bacterial diagnostics without the need for an external power source, making them suitable for implantable diagnostic devices and smart medical systems.

One example is Panda et al.’s development of a TENG-based biodegradable sensor system for detecting E. coli (Figure 7a) [110]. This system combines D-mannose-functionalized 3D-printed polylactic acid (PLA) material with brush-painted silver electrodes to capture bacteria through a simple carbohydrate–protein interaction mechanism. As the concentration of bacteria changes, the sensor’s adsorption capacity varies, leading to changes in resistance, enabling the detection of bacterial concentrations. Additionally, the system was tested on real water samples and unpasteurized milk, confirming its performance. The sensor’s biodegradability was also tested under composting conditions, demonstrating environmental sustainability. This study provides a viable solution for fast and efficient bacterial detection.

Another example is the work by Wang et al., who developed a self-powered TENG-based biosensing system for the detection of Gram-positive bacteria (Figure 7b) [111]. This system employs a vertical contact–separation TENG as a stable voltage source, using vancomycin to capture bacterial cells via specific interactions with the cell walls of Gram-positive bacteria. Additionally, guanidine-functionalized multi-walled carbon nanotubes (CNT-Arg) serve as signal amplification materials to enhance electrical conductivity and bacterial adhesion. By measuring voltage changes, this system precisely detects bacterial concentrations, demonstrating good linearity and low detection limits. Furthermore, the researchers developed a Labview warning program to convert voltage signals into visual alerts, simplifying monitoring. This self-powered system is simple, safe, and environmentally friendly, with potential applications in environmental pollution detection, healthcare, and microbial corrosion.

Overall, using TENGs for pathogen detection holds great potential for advancing healthcare by offering a rapid, sensitive, and cost-effective method for detecting bacterial infections.

#### 3.4.2. Long-Lasting Antibacterials

TENG devices can inhibit bacterial growth and biofilm formation by continuously generating negative charges, offering an innovative antibacterial approach for medical implants and surface disinfection. Compared to traditional antibacterial methods, TENG-based sterilization can effectively avoid antibiotic resistance and the side effects of new antimicrobial materials, while not affecting bone cell adhesion and growth, ensuring long-term stability and safety. This makes it suitable for medical fields such as surgical implants and wound dressings.

For example, Shi et al. proposed a self-powered technology using TENG to provide negative charges for anodized titanium implants (Figure 7c) [112]. This technology converts mechanical energy from human daily movements into electrical energy, providing stable and long-lasting negative charges on the implant surface, effectively inhibiting bacterial adhesion, reducing bacterial counts, and lowering the ratio of live bacteria in mature biofilms. Compared to traditional antibacterial methods, this approach avoids the side effects of new antibacterial components and does not inhibit bone cell adhesion and osteogenesis. This innovative and sustainable antibacterial and bone regeneration method is promising for the future development of orthopedic and dental implants.

Lin et al. developed a flexible, paper-based triboelectric nanogenerator (P-TENG) using ZnO@paper paired with PTFE film (Figure 7d) [113]. This P-TENG demonstrated not only high output performance but also antibacterial activity against E. coli and S. aureus, suggesting applications in surgery for inhibiting and killing bacteria. The study highlighted that ZnO enhances the surface roughness of cellulose paper, thereby improving the output performance of the flexible P-TENG. Moreover, the P-TENG-based pressure sensor shows promise for measuring human motion information.

To conclude, researchers have enabled BM-TENGs to continuously generate negative charges, providing an innovative antibacterial solution for biomaterials. This method effectively suppresses bacterial growth, avoiding antibiotic resistance and the drawbacks of new antimicrobial materials while maintaining bone cell adhesion and regeneration. As a result, it holds great promise in biomaterial applications such as surgical implants and wound dressings, with the potential to greatly enhance the long-term safety and biocompatibility of medical devices.

#### 3.4.3. Mite Removal

Biomaterials such as CS and hydroxyethyl cellulose, with their excellent biocompatibility and mite-resistant properties, are ideal choices for designing anti-mite TENGs. The charges generated through the triboelectric effect can effectively disrupt the growth environment of mites, and, combined with the structural optimization and surface modification of the materials, the anti-mite effects of TENGs can be further enhanced. Such TENG devices can be applied in household items, mattresses, and air purification systems to continuously inhibit mite reproduction.

For instance, Lu et al. synthesized a CTS–hydroxyethyl cellulose pectin (CHP) film via a hydrothermal method, employing it as a triboelectric material [114]. The TENG, containing 11% CTS content, exhibited excellent current and voltage output, high energy conversion efficiency, and anti-mite antibacterial properties. These characteristics highlight its applications in TENG-based smart medical and health monitoring systems.

Overall, the design of self-powered systems for bacterial diagnostics and sterilization enhances both real-time detection and sensitivity while broadening their application range. By optimizing material selection and technological implementation, TENG systems demonstrate significant advantages in bacterial monitoring and antimicrobial treatment, providing innovative solutions for healthcare and environmental management (Table 2).

## 4. Conclusions and Prospects

Compared with traditional polymers, biomaterials not only offer advantages such as non-toxicity, biocompatibility, and biodegradability but they are also abundant in resources and often have a higher surface area, which significantly impacts the output performance of TENG systems. This characteristic provides biomaterials with great potential for various applications.

This paper demonstrates that some BM-TENGs already rival the performance of existing inorganic-based TENGs, showing great promise for self-powered electronic devices, especially in medical diagnostics and health monitoring. With continuous technological advancements, BM-TENGs are expected to find applications in even more fields. These devices can efficiently harness mechanical energy from natural or biological sources, providing both improved user experiences and continuous and reliable self-powered operation, making them particularly suitable for disease diagnosis, bacterial detection, and environmental control.

Although BM-TENG technology faces several challenges, such as limited output performance, uncontrolled degradation, and poor durability, these issues are anticipated to be progressively addressed through meticulous material selection, bionic design, structural optimization, chemical surface modification, and advancements in energy management systems. Furthermore, innovations in surface treatment techniques, enhanced composite materials, and encapsulation technologies will further improve the long-term stability and durability of the devices.

For practical applications of BP-TENG technology in energy harvesting, medical diagnostics, and environmental monitoring, several key issues, such as the energy output, miniaturization, sensitivity, sustainability, and biocompatibility of biomaterials should be addressed.

### 4.1. Energy Output

The energy output of BM-TENGs is often constrained and may not suffice for the high energy demands of certain medical devices. Integrating additional energy harvesting technologies, such as solar and thermoelectric energy collection, can diversify energy sources and augment the overall energy supply. Selecting the appropriate energy harvesting technologies for specific application scenarios is critical to meeting energy needs [115]. Furthermore, developing more efficient energy conversion materials and optimizing device designs are effective ways to enhance energy output. With technological advancements, future BM-TENGs may achieve higher energy densities, thereby better serving various medical applications. Ultimately, the integration of multiple energy harvesting methods will provide strong support for the sustainable development of BM-TENGs.

### 4.2. Biocompatibility and Degradability

Wearable bioelectronic devices must be tissue-compatible to prevent allergic reactions or other adverse effects from direct contact with the human body. Factors influencing degradation rates include chemical composition, structural design, and external environmental conditions, making the degradation process complex and difficult to predict. Additionally, the degradation byproducts must be non-toxic and harmless to prevent negative impacts on human health or the environment [116]. Balancing these factors to ensure efficient performance while safely degrading at the end of its life cycle remains a major challenge for BM-TENG technology in medical and environmental applications.

### 4.3. Comfort and Durability

Considering the clinical characteristics of cardiovascular and other diseases (long-term and sudden occurrences), BM-TENGs need to be worn for extended periods to continuously monitor human health. However, some TENG devices lack breathability and aesthetics, making them unsuitable for long-term wear. Future wearable TENGs should focus more on comfort. By improving the material and microstructure of TENGs, breathability, softness, and stretchability can be enhanced, preventing discomfort and minimizing interference with daily activities [117]. Additionally, BM-TENGs with poor durability can suffer from fatigue and degradation after stretching and deformation [118]. Long-term cyclic testing and simulations should be conducted to evaluate performance changes under repeated stretching.

### 4.4. Integration and Size

To meet the demands of wearable and implantable BM-TENGs, devices must be miniaturized without compromising output performance, it is essential to miniaturize these devices while ensuring that their output performance is not compromised. This necessitates innovative approaches that integrate cutting-edge material science and advanced manufacturing technologies [119]. These advancements offer promising solutions that can enhance the efficiency and functionality of BM-TENGs in compact formats. Addressing the various challenges associated with miniaturization and the integration of different functional units will be a significant focus for future research and development. The ability to effectively combine multiple components into a smaller footprint will not only improve the practicality of these devices in real-world applications but also pave the way for more sophisticated designs that can cater to diverse medical and technological needs. Ultimately, the evolution of BM-TENGs will hinge on the successful integration of these innovations to create highly efficient, adaptable, and user-friendly energy harvesting solutions.

### 4.5. Sensitivity and Stability

The performance of BM-TENGs can be affected by the complexity of biological environments, impacting detection accuracy. For example, low bacterial counts or subtle environmental changes may be difficult for traditional sensors to detect. Therefore, optimizing the structural design and material selection to improve sensitivity in detecting diseases and bacteria is crucial. Researchers can introduce micro-nanostructures, such as wrinkles, hemispherical arrays, pyramid arrays, and nanowires, which can increase the surface charge density of TENGs and enhance contact with detected substances, thereby improving sensitivity. In terms of material selection, methods such as doping, injection, and coating with exogenous chemicals can optimize the surface charge affinity of TENGs, increasing output voltage and improving detection performance. Additionally, assessing long-term stability under various environmental conditions is equally important, as it ensures device reliability across diverse applications. Continuous technological innovation and materials research will drive the application prospects of BM-TENGs in the biomedical field. Ultimately, combining multiple approaches will help achieve higher sensitivity and stability to meet future medical demands.

In conclusion, by addressing these technical challenges, BM-TENGs hold the promise of advancing clinical applications, achieving efficient and stable energy harvesting, and providing strong support for future health monitoring and sustainable development.

## Figures and Tables

**Figure 1 nanomaterials-14-01885-f001:**
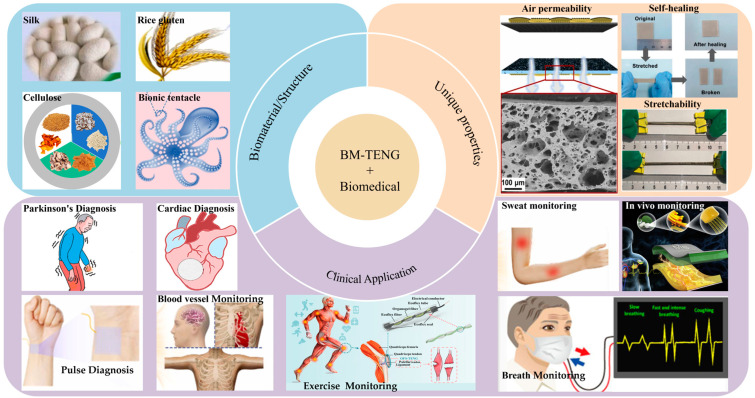
Recent progress of TENG based on biomaterials including biomaterials like silk; reprinted with permission from Ref. [53]. Copyright 2023, Elsevier; rice gluten and its application as blood vessel and sweat monitoring; reprinted with permission from Ref. [54]. Copyright 2022, Elsevier; Cellulose; reprinted with permission from Ref. [55]. Copyright 2022, Elsevier; Bionic tentacle; reprinted with permission from Ref. [56]. Copyright 2024, Elsevier; Air permeability; reprinted with permission from Ref. [57]. Copyright 2022, Elsevier; Self-healing; reprinted with permission from Ref. [58]. Copyright 2021, Elsevier; Stretchability; reprinted with permission from Ref. [59]. Copyright 2021, Wiley Online Library; Real-time monitoring of Parkinson’s disease; reprinted with permission from Ref. [60]. Copyright 2021, Elsevier; Cardiac diagnosis; reprinted with permission from Ref. [61]. Copyright 2023, ACS Publications; Pulse diagnosis; reprinted with permission from Ref. [62]. Copyright 2023, Elsevier; Exercise Monitoring; reprinted with permission from Ref. [63]. Copyright 2022, ACS Publications; In vivo monitoring; reprinted with permission from Ref. [64]. Copyright 2020, Elsevier; Breath monitoring; reprinted with permission from Ref. [65]. Copyright 2020, ACS Publications.

**Figure 2 nanomaterials-14-01885-f002:**
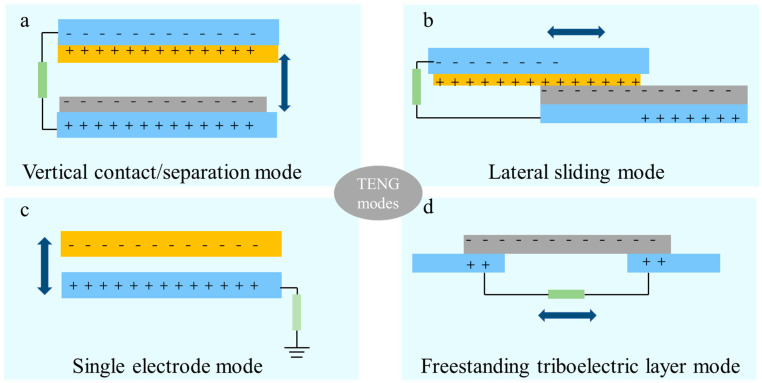
The four primary working modes of TENG are (**a**) vertical contact–separation mode, (**b**) lateral-sliding mode, (**c**) single-electrode mode, and (**d**) freestanding triboelectric-layer mode.

**Figure 3 nanomaterials-14-01885-f003:**
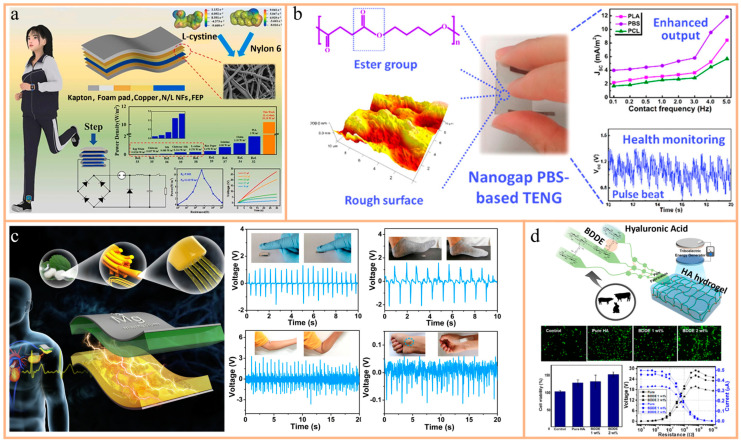
(**a**) Structure of HM-TENG: the upper part represents the simulation and SEM of L-cystine and the original nylon 6; the lower part represents the power density of the HM-TENG at different external resistances, with the charge curve of different capacitors. Reprinted with permission from Ref. [90]. Copyright 2023, Elsevier. (**b**) Structure and output of nano-gap TENG: right shows its output current density and the voltage signal used to monitor the pulse. Reprinted with permission from Ref. [91]. Copyright 2020, ACS Publications. (**c**) Structure design of SNR-TENG and its application in medical monitoring. Reprinted with permission from Ref. [64]. Copyright 2020, Elsevier. (**d**) Top shows the preparation process of HA, middle shows the results of the cell viability assay of MC3T3-E1 after being treated with HA membrane for 24 h, and bottom shows the triboelectric outputs of the TENG based on HA and PTFE films under various external resistances. Reprinted with permission from Ref. [92]. Copyright 2020, Elsevier.

**Figure 4 nanomaterials-14-01885-f004:**
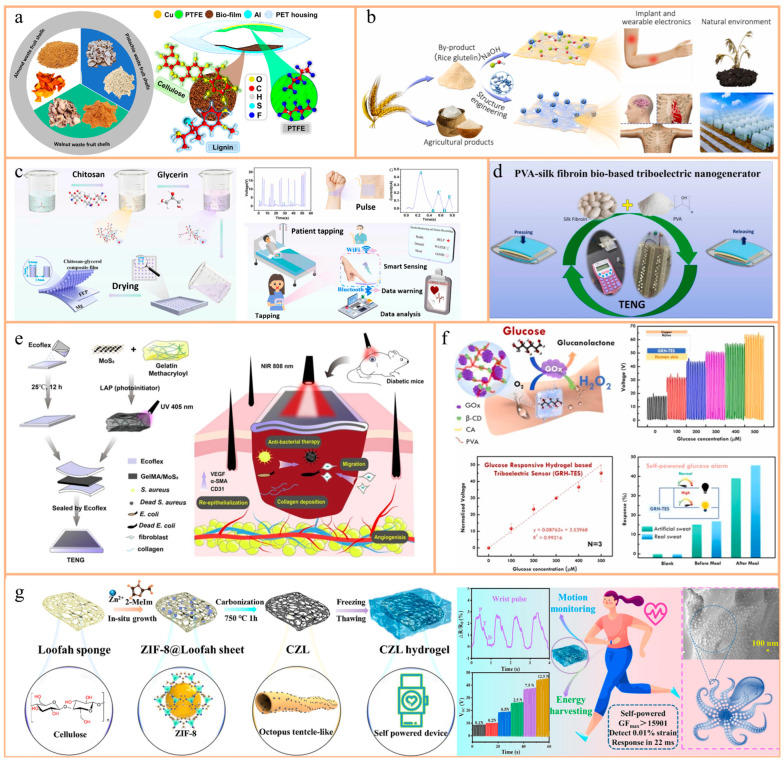
(**a**) Source of wood fiber and schematic diagram of the TENG device based on wood fiber substrate. Reprinted with permission from Ref. [55]. Copyright 2022, Elsevier. (**b**) Rice gluten films were prepared by the traditional method using NaOH (as shown in the upper diagram) and then fabricated into biocompatible and environmentally friendly triboelectric materials according to structural engineering (as shown in the lower diagram). Reprinted with permission from Ref. [54]. Copyright 2022, Elsevier. (**c**) Preparation of CS–glycerin composite film, its output performance as a TENG, and its application in medical health. Reprinted with permission from Ref. [62]. Copyright 2023, Elsevier. (**d**) Schematic of the structure of the PVA/SF-based TENG. Reprinted with permission from Ref. [53]. Copyright 2023, Elsevier. (**e**) Schematic of the MoS2-based TENG patch for accelerating wound healing. Reprinted with permission from Ref. [96]. Copyright 2024, Elsevier. (**f**) Schematic of the GAH-TES for sweat monitoring. TENG voltage response when detecting 0–500 μM glucose in artificial sweat and calibration curve for repeated glucose detection. Reprinted with permission from Ref. [97]. Copyright 2023, Elsevier. (**g**) Schematic diagrams of the structure of CZL hydrogel, SEM (Scanning Electron Microscopy), pulse monitoring, and energy harvesting. Reprinted with permission from Ref. [56]. Copyright 2024, Elsevier.

**Figure 5 nanomaterials-14-01885-f005:**
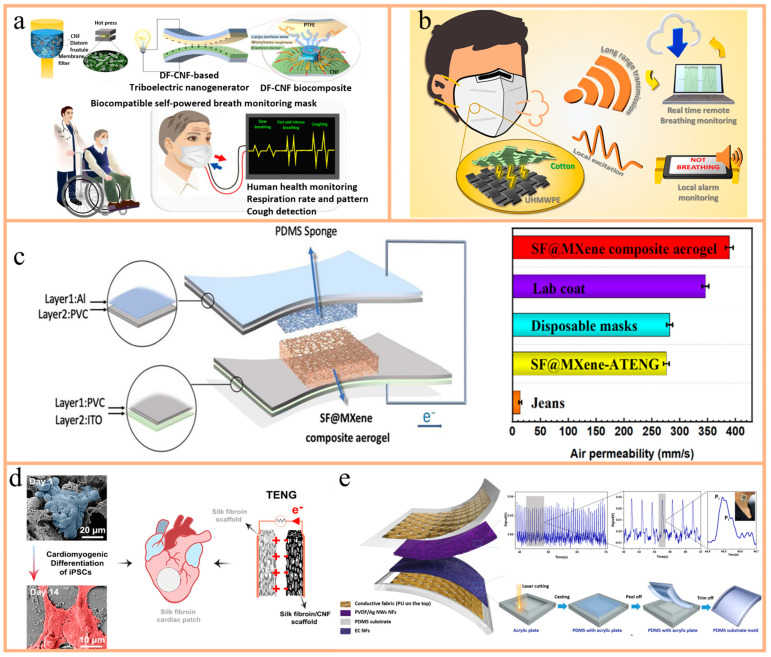
(**a**) Schematic diagrams of the manufacturing process of DF-CNF-based TENG and the self-powered biocompatible smart mask for human respiratory sensing and health monitoring. Reprinted with permission from Ref. [65]. Copyright 2020, ACS Publications. (**b**) AF-TENG fabricated mask for patient respiratory detection via Wi-Fi and LoRa. Reprinted with permission from Ref. [103]. Copyright 2023, ACS Publications. (**c**) Left shows the structure of SF@MXene-TENG and right shows the air permeability of common textile materials and SF@MXene-TENG as well as their corresponding error bars. Structure of SF@MXene-TENG mask fabricated for diagnosing asthma symptoms. Reprinted with permission from Ref. [104]. Copyright 2023, ACS Publications. (**d**) SF/CNF scaffold for cardiac motion energy and iPSCs images captured over 14 days. Reprinted with permission from Ref. [61]. Copyright 2023, ACS Publications. (**e**) Manufacturing process and structure of NFM-TENG and respiratory monitoring. Reprinted with permission from Ref. [105]. Copyright 2019, ACS Publications.

**Figure 6 nanomaterials-14-01885-f006:**
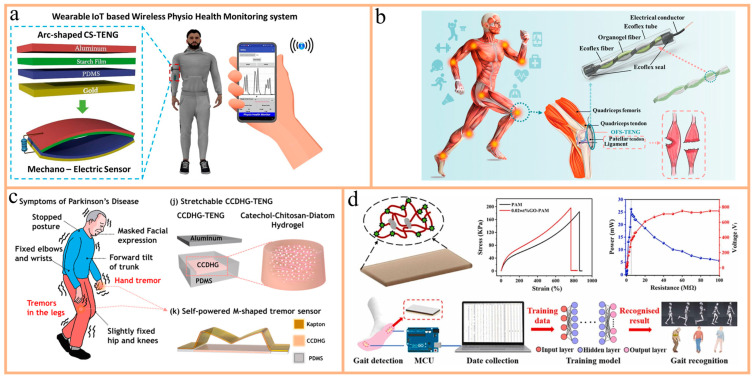
(**a**) Schematic diagram of WPHM connected to the human body via arc-shaped CS-TENG and its wireless monitoring application. Reprinted with permission from Ref. [108]. Copyright 2024, Elsevier. (**b**) Concept and structural schematic of OFS-TENG implanted in the human body. Reprinted with permission from Ref. [63]. Copyright 2022, ACS Publications. (**c**) Structural schematic of CCDHG-TENG and real-time monitoring of typical symptoms of Parkinson’s disease. Reprinted with permission from Ref. [60]. Copyright 2021, Elsevier. (**d**) Self-powered strain sensor based on GO-PAM hydrogel for monitoring human motion. Reprinted with permission from Ref. [109]. Copyright 2022, Elsevier.

**Figure 7 nanomaterials-14-01885-f007:**
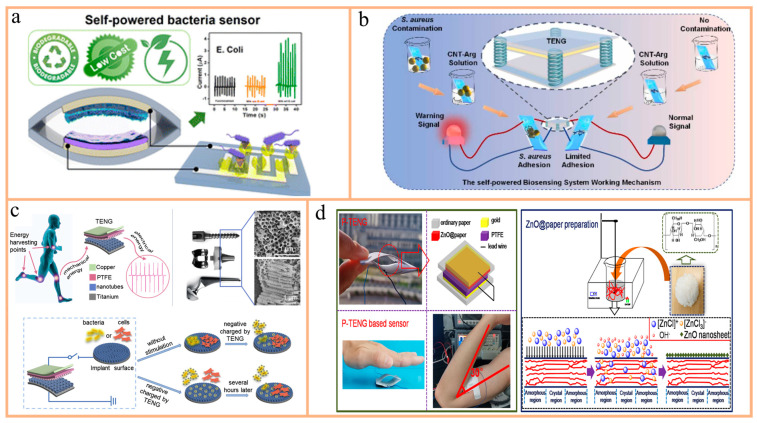
(**a**) Schematic diagrams of the structure of the PLA-TENG self-powered sensor system and the current signals obtained by monitoring the concentration of bacteria. Reprinted with permission from Ref. [110]. Copyright 2023, ACS Publications. (**b**) CNT-Arg-based TENG measures Gram-positive bacteria through voltage measurement. Reprinted with permission from Ref. [111]. Copyright 2022, Elsevier. (**c**) Top shows the structure of TENG and the surface morphology of Ti orthopedic implants after anodization. Bottom shows the mechanism of action in the antibacterial and osteogenesis-promoting surfaces of the TENG. Reprinted with permission from Ref. [112]. Copyright 2020, Elsevier. (**d**) Left shows the structure and working principle of P-TENG and its function as an antibacterial pressure sensor for monitoring human motions; right shows the preparation process of ZnO@paper. Schematic of antimicrobial pressure sensor based on P-TENG for monitoring human motion. Reprinted with permission from Ref. [113]. Copyright 2021, MDPI.

**Table 1 nanomaterials-14-01885-t001:** Summary of TENGs with different types of biomaterials used in the biomedical field.

Material	Type	Size	Output	Energy Source	Ref.
Lignin	Natural material	4.5 × 4.5 cm^2^	700 V/95 μA	Hand slapping	[55]
Rice glutelin	Natural material	6 cm diameter	170 V/14 μA	Oscillating force	[54]
Fish gelatin	Natural material	3 × 3 cm^2^	500 V	Joint movement	[98]
CS/glycerol	Biocomposites	0.6 × 2 × 1.5 mm^3^	127 V	Pulse	[62]
PVA/silk	Biocomposites	4 × 7 cm^2^	172 V/8.5 μA	Oscillating force	[53]
MoS_2_/gelatin-methacryloyl	Hydrogel	2 × 2 × 1 mm^3^	48.8 V/0.57 μA	Finger curvature	[96]
β-Cyclodextrin/PVA	Hydrogel	2 × 2 cm^2^	65 V/3.5 μA	Movement	[97]

**Table 2 nanomaterials-14-01885-t002:** Summary of the performance and application of BP-TENG with different biomaterials.

Material	Position	Size	Electrical Output	Application	Ref.
CNF/DFs	Mouth	35 × 25 × 12 mm^3^	388 V/18.6 μA	Respiratory diagnosis	[65]
Cotton/UHMWPE	Mouth	4 × 4 cm^2^	∼20 V	Respiratory diagnosis	[103]
SF/MXene	Mouth	2 × 2 cm^2^	545 V/16.13 μA	Asthma diagnosis	[104]
SF/CNF	Cardiomyogenic	1 × 5 cm^2^	0.46 V/4.5 nA	Cardiac diagnosis	[61]
ZIF-8@loofah	Elbow	—	54.2 V/309 nA	Pulse diagnosis	[56]
Cornstarch	Hand/Feet	4 × 3 cm^2^	266 V/0.49 μA	Exercise monitoring	[108]
Organogel/silicone	Ligament	1.8 × 0.5 mm^2^ × 3 cm	0.7 V	Exercise monitoring	[63]
GO-PAM	Feet	5 × 5 cm^2^ × 1 mm	990 V/63.84 μA	Parkinson’s diagnosis	[109]
D-mannose/PLA	Water/milk	2 × 2 cm^2^	70 V/800 nA	Bacterial diagnostics	[110]
Titanium oxide	Bone/tooth	1.6 × 1 cm^2^	12 V/0.15 μA	Antibacterial	[112]
ZnO@paper	Hand	6 × 6 cm^2^	77 V/0.17 μA	Antibacterial	[113]

a “—” in the table indicates that the data were not documented in the research.

## Data Availability

Data available upon request from the authors.

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
