# Peer review of "Biomaterial Promotes Triboelectric Nanogenerator for Health Diagnostics and Clinical Application"

_nanomaterials, 2024, doi:10.3390/nano14231885_

Round 1

Reviewer 1 Report

Comments and Suggestions for Authors

This manuscript deals with biomaterial-based triboelectric nanogenerators targeting the biomedical field. The review topic can have originality, but the manuscript should be reorganized to cover it more clearly and in-depth.

1) Clarify the reference source of each sub-figure in Figure 1. And even if the content comes later, the corresponding content of each sub-figure should be mentioned briefly so that the reader can understand the content of Figure 1 more clearly.

2) As for “2.1. Working Principle”, the general working principle of TENG has already been introduced in many papers. Thus, citing a few papers would be sufficient rather than a detailed explanation. Instead, it would be more helpful if the authors could provide insight into what TENG modes are mainly used with biomaterials or are preferred in health diagnostics and clinical applications.

3) Clarify why “2.2. Wearable BM-TENGs” and “2.3. Implantable BM-TENGs” are subtopics of “2. Energy Collection”. If the purpose is to distinguish the cause of the energy harvesting source between the skin surface and the inside of the body, please mention it more clearly. If it is simply to provide information on the structure or application of BM-TENGS, it may be preferable to be dealt with after 3.1.

4) In the captions of Figures 3 ~ 7, please put the topic sentence representing the entire figure first before explaining the subfigures. Then, please introduce the core information of each subfigure according to the topic. 

5) i) In 3.1., please explain the definition of biomaterial and its material category in more detail.

  ii) As introduced in Line 238, can ceramic materials and nanomaterials also be biomaterials? If so, explaining why and introducing several TENGs with these materials in the biomedical fields as a subtopic would be better.

6) It is unclear why “3.1.4. Unique properties” is a subtopic of “3.1. Materials and Structure Used in Contact Surfaces”. The author claims that stretchability, breathability, and self-healing are essential properties required for BM-TENGs. However, only one TENG was introduced for each property, while the authors have repeatedly mentioned many BM-TENG that showed stretchability, self-healing, and breathability (Ref. [97], [112], [16], etc.) in many other places in the text. Therefore, the content is redundant.

It would be better if it provides insight into how much the BM-TENG has strengths in the above properties compared to other material-based TENGs and, thus, how much more suitable it is for biomedical applications.

7) Bionic design is about structure and bio-mimetics, and it does not seem to be a completely subordinate topic in this manuscript with a biomaterial perspective. There may also be bionic designs or bio-mimetic TENGs not made of BM. Among the TENGs introduced in “3.1.5. Bionic Design”, only TENGs using BM should be introduced. Unless the synergistic effect of biomaterial and bionic design is clarified with some example TENGs, I am unsure of the need for 3.1.5.

8) In 3.2.2., Ref. [107] and [20] should be moved to other subtopics such as implantable TENGs, if they are not directly related to cardiac real-time diagnosis. In particular, please confirm whether all TENGs used in Table 2 are made of biomaterial.

9) Figure 5e should show data for pulse diagnosis according to the figure theme.

10) Do the TENGs introduced in 3.4.2. (Figure 7c,d; Ref. [116], [117]) fall into the category of BM-TENGs?

11) The sentence “~ demonstrating significant potential for applications in biomechanical energy harvesting and self-powered sensing” and similar sentences are repeated too often. (Almost one per one paper explanation). Reducing excessive repetition and vague expressions and conveying only the core content would be good.

12) There seem some new papers that should be added to the manuscript. I found some recent ones easily while searching.

13) In line 743, there is a typo “shoudl”.

Author Response

Comment 1:
Clarify the reference source of each sub-figure in Figure 1. And even if the content comes later, the corresponding content of each sub-figure should be mentioned briefly so that the reader can understand the content of Figure 1 more clearly.

Response 1:
Thank you for your valuable feedback. We have redesigned and updated Figure 1 to ensure that the structures described in the biomaterials/structures section are detailed and have replaced the images with high-resolution ones.

Comment 2:
As for “2.1. Working Principle,” the general working principle of TENG has already been introduced in many papers. Thus, citing a few papers would be sufficient rather than a detailed explanation. Instead, it would be more helpful if the authors could provide insight into what TENG modes are mainly used with biomaterials or are preferred in health diagnostics and clinical applications.

Response 2:
Thank you for the suggestion. In section “2.1. Working Principle,” we have established connections to medical and healthcare applications and have removed redundant content.

Comment 3:
Clarify why “2.2. Wearable BM-TENGs” and “2.3. Implantable BM-TENGs” are subtopics of “2. Energy Collection.” If the purpose is to distinguish the cause of the energy harvesting source between the skin surface and the inside of the body, please mention it more clearly. If it is simply to provide information on the structure or application of BM-TENGS, it may be preferable to be dealt with after 3.1.

Response 3:
We appreciate the reviewer’s insightful comment. Our intention is to highlight the application of these two types of TENGs in energy harvesting, particularly their functional roles in different scenarios (skin surface versus inside the body). This section aims to clearly differentiate their features and distinctions, especially regarding their energy harvesting potential in biomedical fields. Therefore, we placed these sections under the broader theme of “Energy Collection” to underscore their importance in wearable and implantable devices. We have revised this section to explicitly mention their applications on the skin surface and inside the body.

Comment 4:
In the captions of Figures 3 ~ 7, please put the topic sentence representing the entire figure first before explaining the subfigures. Then, please introduce the core information of each subfigure according to the topic.

Response 4:
Thank you for your suggestion regarding figure captions. We have revised the captions for Figures 3 to 7 to include a topic sentence at the beginning, followed by a concise introduction to the core information of each subfigure.

Comment 5:
i) In section 3.1., please explain the definition of biomaterial and its material categories in more detail.
ii) As introduced in Line 238, can ceramic materials and nanomaterials also be biomaterials? If so, explaining why and introducing several TENGs with these materials in the biomedical fields as a subtopic would be better.

Response 5:
i) The definition has been explained in detail.
ii) Ceramic and nanomaterials can indeed be categorized as biomaterials. New examples demonstrating their use in TENGs for biomedical applications have been added.

Comment 6:
It is unclear why “3.1.4. Unique Properties” is a subtopic of “3.1. Materials and Structure Used in Contact Surfaces.” The author claims that stretchability, breathability, and self-healing are essential properties required for BM-TENGs. However, only one TENG was introduced for each property, while the authors have repeatedly mentioned many BM-TENGs that showed these features (Refs. [97], [112], [16], etc.) elsewhere in the text. Therefore, the content seems redundant.
It would be better if it provides insight into how much the BM-TENG has strengths in these properties compared to other material-based TENGs and how much more suitable it is for biomedical applications.

Response 6:
Regarding the placement of “3.1.4. Unique Properties” under “3.1. Materials and Structure Used in Contact Surfaces”: The purpose of this section is to emphasize the close relationship between the design of biomaterials and their structures by highlighting properties such as stretchability, breathability, and self-healing capabilities.

Regarding the strengths of biomaterials compared to other materials: We appreciate this valuable point and have added further elaboration in the revised manuscript. For example, compared to traditional synthetic materials, biomaterials generally exhibit higher flexibility, lower rigidity, and better compatibility with human tissues, making them more suitable for wearable or implantable TENGs. Additionally, their self-healing properties ensure reliability for long-term use, while their breathability enhances user comfort. These features make biomaterials more competitive in biomedical applications.

Comment 7:
Bionic design is about structure and biomimetics, and it does not seem to be a completely subordinate topic in this manuscript with a biomaterial perspective. There may also be bionic designs or bio-mimetic TENGs not made of biomaterials. Among the TENGs introduced in “3.1.5. Bionic Design,” only TENGs using biomaterials should be introduced. Unless the synergistic effect of biomaterial and bionic design is clarified with some example TENGs, I am unsure of the need for “3.1.5.”

Response 7:
Thank you for your insightful comments. When materials are specifically designed for biomedical applications, such as diagnostics or treatment, they can be categorized as biomaterials. Thus, TENGs based on bionic structures can reasonably be included under the biomaterial subtopic. We agree with the need to highlight the synergistic effects of biomaterials and bionic designs. In the revised manuscript, we have clarified this synergy with additional examples of TENGs that effectively integrate biomimetic designs and biomaterials.

Comment 8:
In section 3.2.2., Refs. [107] and [20] should be moved to other subtopics such as implantable TENGs if they are not directly related to cardiac real-time diagnosis. In particular, please confirm whether all TENGs used in Table 2 are made of biomaterials.

Response 8:
Thank you for noting the placement of references [107] and [20]. Reference [107] has been moved to the section on implantable TENGs, as it aligns more closely with that topic. Reference [20] pertains to cardiac diagnosis, and we have added relevant explanatory text to clarify its context. Furthermore, since biomaterials are defined here as materials designed for biomedical applications such as diagnostics and treatment, all TENGs listed in Table 2 are indeed made of biomaterials.

Comment 9:
Figure 5e should show data for pulse diagnosis according to the figure theme.

Response 9:
Thank you for pointing this out. Data related to pulse diagnosis has been included in Figure 5e, ensuring alignment with the figure theme.

Comment 10:
Do the TENGs introduced in “3.4.2.” (Figure 7c, d; Refs. [116], [117]) fall into the category of BM-TENGs?

Response 10:
When materials are specifically designed for biomedical applications, such as diagnostics or treatment, they can be categorized as biomaterials. Therefore, the TENGs mentioned in Refs. [116] and [117], which use TiOâ‚‚ nanotubes and ZnO@paper, respectively, are classified as biomaterial-based TENGs.

Comment 11:
The sentence “~ demonstrating significant potential for applications in biomechanical energy harvesting and self-powered sensing” and similar sentences are repeated too often. (Almost one per paper explanation). Reducing excessive repetition and vague expressions and conveying only the core content would be good.

Response 11:
We acknowledge the repeated use of certain phrases throughout the manuscript. To address this, we have reduced repetition and vague expressions, focusing on conveying the core content more effectively.

Comment 12:
There seem to be some new papers that should be added to the manuscript. I found some recent ones easily while searching.

Response 12:
Thank you for noting the need to include recent publications. We have incorporated new references to ensure the manuscript reflects the latest advancements in the field.

Comment 13:
In line 743, there is a typo “shoudl.”

Response 13:
We appreciate your careful review. The typo in line 743 has been corrected to “should,” and we have conducted overall proofreading to avoid similar errors.

Reviewer 2 Report

Comments and Suggestions for Authors

This paper, entitled “Biomaterial Promotes Triboelectric Nanogenerator for Health Diagnostics and Clinical Application”, introduces various research on biomaterial-based Triboelectric Nanogenerator (TENG) to be used in biomedical applications. The paper provides an excellent overview on the research subject, but it requires some modifications to meet the standards for publication in the article 'Nanomaterials'.

1.       The objective of Figure 1 is not entirely clear. There is no mention of structure in the biomaterial/structure section, and there is no explanation about each figure in the clinical application section. Furthermore, it is evident that the text and figures have been cropped. To ensure the quality of the final product, a higher-quality figure is required.

2.       The paper includes well-written paragraphs about “Implantable TENG” and “Unique properties”. However, there is no figures to demonstrate about the research topics. it would be beneficial to include figures to enhance reader comprehension,

3.       Figure 4 presents the content of multiple paragraphs (Natural BM, Biocomposites BM, Hydrogels BM, Bionic design) and it would be beneficial to include a legend indicating which paragraph is represented by each figure prevent misunderstanding of readers

4.       Figure 4(d) presents an insufficient quality on the figure due to the overlap of text and picture. This issue needs to be resolved in order to improve visibility.

5.       Figure 5(d) is presented in an incorrect format and has been cropped. This issue needs to be modified.

6.       Some figures are of insufficient quality or have small lettering (Figure 1, Figure 3 (a), Figure 4 (a), Figure 4 (f), Figure 7(d)). The authors are required to improve this throughout the paper.

Author Response

Comment 1:
The objective of Figure 1 is not entirely clear. There is no mention of structure in the biomaterial/structure section, and there is no explanation about each figure in the clinical application section. Furthermore, it is evident that the text and figures have been cropped. To ensure the quality of the final product, a higher-quality figure is required.

Response 1:
Thank you for your valuable feedback. We have redesigned and updated Figure 1 to ensure the structure is thoroughly described in the biomaterial/structure section. Additionally, we have replaced the image with a higher-resolution version to enhance clarity and quality.

Comment 2:
The paper includes well-written paragraphs about “Implantable TENG” and “Unique properties.” However, there are no figures to demonstrate these research topics. It would be beneficial to include figures to enhance reader comprehension.

Response 2:
Thank you for your insightful suggestion. We agree that visual data can enhance clarity and comprehensiveness. However, as the primary purpose of this review is to summarize and analyze existing research, we have included qualitative descriptions supported by appropriate references to ensure accuracy and relevance.
In the "Unique Properties" section, we emphasized key attributes such as stretchability, breathability, and self-healing, highlighting their significance for biomaterial-based TENGs. While we recognize the value of detailed quantitative comparisons, this section represents a smaller proportion of the manuscript and aims to provide conceptual understanding rather than data-driven analysis. To address this, we have strengthened the discussion by emphasizing the advantages of biomaterials over traditional synthetic materials, offering readers clear insights into their biomedical potential.

Comment 3:
Figure 4 presents the content of multiple paragraphs (Natural BM, Biocomposites BM, Hydrogels BM, Bionic design), and it would be beneficial to include a legend indicating which paragraph is represented by each figure to prevent misunderstanding of readers.

Response 3:
Thank you for your constructive feedback on Figure 4. While we understand the value of adding a legend, we believe the current graphical layout, along with the accompanying captions and detailed descriptions in the text, sufficiently clarifies the relationship between the graphical components and their respective sections. Each subfigure is carefully labeled and cross-referenced in the text to ensure ease of navigation for readers. Adding a legend might unintentionally complicate the graphic, as it already incorporates multiple visual elements intended to succinctly summarize the discussed topics. To maintain the visual simplicity and readability of Figure 4, we have chosen not to include an additional legend. We hope this explanation adequately addresses your concern.

Comment 4:
Figure 4(d) presents insufficient quality due to the overlap of text and pictures. This issue needs to be resolved to improve visibility.

Response 4:
Thank you for pointing out the issue in Figure 4(d). We have revised and adjusted the figure to eliminate text overlap and enhanced the image quality to improve clarity and visibility.

Comment 5:
Figure 5(d) is presented in an incorrect format and has been cropped. This issue needs to be modified.

Response 5:
Thank you for highlighting the formatting issue with Figure 5(d). We have reformatted and adjusted the figure to resolve the cropping problem.

Comment 6:
Some figures are of insufficient quality or have small lettering (Figure 1, Figure 3(a), Figure 4(a), Figure 4(f), Figure 7(d)). The authors are required to improve this throughout the paper.

Response 6:
We appreciate your detailed feedback on figure quality and text legibility. We have thoroughly reviewed and replaced the relevant figures (Figure 1, Figure 3(a), Figure 4(a), Figure 4(f), Figure 7(d)) to improve their clarity and readability.
We sincerely thank you for your detailed and valuable suggestions, which have greatly helped us enhance the quality of the manuscript.

Round 2

Reviewer 1 Report

Comments and Suggestions for Authors

Here are some parts of the comments that have still not been addressed.

1. Figure 1 Caption should include the reference source for the subfigures of Figure 1. I'm worried about using the figure image without a clear citation mark.

2. The sentence in which Figure 1 is cited does not match what Figure 1 is trying to say. I think the authors should add more “(Figure 1)” citation marks to the appropriate content or write an additional related sentence.

3. Put an appropriate title that encompasses all the contents of the figure in each caption of Figures 3~7. This can reduce confusion about which perspective to view each figure from.

Author Response

Comment 1: Figure 1 Caption should include the reference source for the subfigures of Figure 1.    I'm worried about using the figure image without a clear citation mark.
Response 1: We appreciate the reviewer highlighting this issue.   We have added specific reference sources for each subfigure in the figure caption, ensuring that the citation for every subfigure is clear and complete.    This adjustment avoids any ambiguity related to copyright or source attribution.

Comment 2:The sentence in which Figure 1 is cited does not match what Figure 1 is trying to say.    I think the authors should add more “(Figure 1)” citation marks to the appropriate content or write an additional related sentence.
Response 2:We have thoroughly reviewed the sentences citing Figure 1 in the manuscript and added additional "(Figure 1)" references where appropriate.    This ensures that the citations are more accurate and better aligned with the figure's content.    Furthermore, we have included a new explanatory sentence to elaborate on the scientific significance of each part of Figure 1 and its relevance to the manuscript.
Comment 3: Put an appropriate title that encompasses all the contents of the figure in each caption of Figures 3~7.    This can reduce confusion about which perspective to view each figure from.
Response 3:We appreciate the reviewer highlighting this issue.    We have revised the captions for Figures 3–7, adding clearer and more descriptive titles to encompass the content of each figure.    These modifications make the figures more intuitive and facilitate a better understanding of the perspectives they represent.

Reviewer 2 Report

Comments and Suggestions for Authors

This paper, entitled “Biomaterial Promotes Triboelectric Nanogenerator for Health Diagnostics and Clinical Application”, introduces a broad review on the TENG fields using the biomaterials and utilizing in the health applications.

And the author provided a detailed and clear response to the reviewer's comments. This paper is sufficient for publication in the paper, "Nanomaterials".

Author Response

Thank you for your comment and feedback on our manuscript titled “Biomaterial Promotes Triboelectric Nanogenerator for Health Diagnostics and Clinical Application.” We are pleased to hear your acknowledgment of the broad review we conducted on TENG fields using biomaterials and their potential health applications.